# Kinetic Sensors for Ligament Balance and Kinematic Evaluation in Anatomic Bi-Cruciate Stabilized Total Knee Arthroplasty

**DOI:** 10.3390/s21165427

**Published:** 2021-08-11

**Authors:** Luigi Sabatini, Francesco Bosco, Luca Barberis, Daniele Camazzola, Alessandro Bistolfi, Salvatore Risitano, Alessandro Massè, Pier Francesco Indelli

**Affiliations:** 1Department of Orthopaedics and Traumatology, University of Torino, Via Zuretti, 29, 10126 Turin, Italy; francescobosco1992@gmail.com (F.B.); l.barberis90@gmail.com (L.B.); daniele.camazzola@gmail.com (D.C.); a.bistolfi@libero.it (A.B.); alessandro.masse@unito.it (A.M.); 2Department of Orthopaedic Surgery and Traumatology, “Maggiore” Hospital of Chieri, Via de Maria, 1, 10023 Turin, Italy; srisitano@gmail.com; 3Department of Orthopaedic Surgery and Bioengineering, Stanford University School of Medicine, Palo Alto Veterans Affairs Health Care System (PAVAHCS), Palo Alto, CA 94304, USA; pindelli@stanford.edu

**Keywords:** kinetic sensor, total knee arthroplasty, knee balance, medial pivot, femoral rollback, predictor variables

## Abstract

Sensor technology was introduced to intraoperatively analyse the differential pressure between the medial and lateral compartments of the knee during primary TKA using a sensor to assess if further balancing procedures are needed to achieve a “balanced” knee. The prognostic role of epidemiological and radiological parameters was also analysed. A consecutive series of 21 patients with primary knee osteoarthritis were enrolled and programmed for TKA in our unit between 1 September 2020 and 31 March 2021. The VERASENSE Knee System (OrthoSensor Inc., Dania Beach, FL, USA) has been proposed as an instrument that quantifies the differential pressure between the compartments of the knee intraoperatively throughout the full range of motion during primary TKA, designed with a J-curve anatomical femoral design and a PS “medially congruent” polyethylene insert. Thirteen patients (61.90%) showed a “balanced” knee, and eight patients (38.10%) showed an intra-operative “unbalanced” knee and required additional procedures. A total of 13 additional balancing procedures were performed. At the end of surgical knee procedures, a quantitatively balanced knee was obtained in all patients. In addition, a correlation was found between the compartment pressure of phase I and phase II at 10° of flexion and higher absolute pressures were found in the medial compartment than in the lateral compartment in each ROM degree investigated. Moreover, those pressure values showed a trend to decrease with the increase in flexion degrees in both compartments. The “Kinetic Tracking” function displays the knee’s dynamic motion through the full ROM to evaluate joint kinetics. The obtained kinetic traces reproduced the knee’s medial pivot and femoral rollback, mimicking natural knee biomechanics. Moreover, we reported a statistically significant correlation between the need for soft tissue or bone resection rebalancing and severity of the initial coronal deformity (>10°) and a preoperative JLCA value >2°. The use of quantitative sensor-guided pressure evaluation during TKA leads to a more reproducible “balanced” knee. The surgeon, evaluating radiological parameters before surgery, may anticipate difficulties in knee balance and require those devices to achieve the desired result objectively.

## 1. Introduction

Over the past decade, total knee arthroplasty (TKA) rates are rising due to the increased life expectancy of the population and the number of osteoarthritis cases found. Therefore, the demand for primary total knee arthroplasties is projected to grow in the following years [1,2]. Despite the recent development of modern technologies such as computer-assisted navigation and additive layer manufacturing, instability following total knee arthroplasty remains one of the main causes of TKA revision. In the literature, revision total knee arthroplasty (r-TKA) due to instability, related to inappropriate soft tissue balancing, has been estimated in more than 20% each year [3]. For this reason, surgeons may tend to increase the intra-articular level of constraint to improve implant stability, resulting in less range of motion, premature wear and reduced total implant survival [4,5,6]. Recent studies in the literature have shown that patients undergoing TKA better appreciate a moderate degree of laxity, especially in the lateral compartment, compared with greater stiffness [7,8]. Nevertheless, a desirable medial pivot kinematic is often not reproduced in TKA described by Dennis et al. [9]. Therefore, research is currently ongoing to find a reproducible method to intraoperatively achieve optimal ligamentous balance and adequate rotational alignment of prosthetic components to improve function and survival of total knee arthroplasties and greater patient satisfaction. To date, there are still no reproducible protocols for achieving ideal balance and stability in primary total knee arthroplasty [10]. Regardless of the procedures used (measured resection, gap-balancing technique, or combined), intraoperatively determining knee stability is extremely surgeon-dependent [4]. This assessment can be achieved differently through first-generation total knee arthroplasty instruments (lamina spreaders, spacer blocks) or more modern tensiometers [11,12,13]. During the last few years, sensor technology has been introduced into knee prosthetics. One of these systems is the VERASENSE Knee System (OrthoSensor Inc., Dania Beach, FL, USA), which, by measuring the medial and lateral compartmental knee pressures, evaluates the kinematics of the TKA and corrects the balance of the operated knee in real time [14]. Several authors [14,15], based on intraoperative observations of experienced surgeons and biomechanical studies [14,15,16], defined knees as adequately “balanced” when the pressure difference between the compartments was less than 15 pound-force (lbf) throughout the entire range of motion. The authors define this digital sensor system as an excellent method to obtain ideal stability during primary total knee arthroplasty. From the data acquired during soft tissue balancing using a sensor, the surgeon could reduce the complications related to ligament balancing in the future [15]. This study aims to quantify the differential pressure between the medial and lateral compartments of the knee throughout the full range of motion during primary Anatomic Bi-cruciate Stabilized (BCS) TKA using a kinetic sensor. In addition, we recorded the additional soft tissue releases or bone resection needed to achieve a “balanced” knee after the traditional balancing procedures were performed. Moreover, the role of epidemiological and radiological parameters was analysed to assess if they might be prognostic factors to define the need to use a sensor to improve TKA balancing.

## 2. Materials and Methods

A consecutive series of 21 BCS Anatomical TKA were implanted in our unit between 1 November 2020 and 31 March 2021. Patients under 60 years of age, with valgus deformity greater than 5°, with rheumatoid arthritis or other immunologically joint, infective, or neurological diseases, patients undergoing revision TKA, previous ligament reconstruction, previous osteotomies or previous severe traumatic surgical treatment around the knee were excluded from the study. Moreover, all patients with missing data were excluded. For each patient, the mechanical alignment of the operating knee was assessed on a weight-bearing x-ray of the entire lower limb prior to surgery, defining varus knees with a hip-knee-ankle (HKA) of −3° or less; valgus knees when an HKA of +3° or more was measured. Furthermore, an anterior-posterior and lateral knee weight-bearing view, a Rosenberg and a Merchant and Lauren view were performed to achieve a better pre-operative radiographic planning [17]. Furthermore, the joint line convergence angle (JLCA) was also investigated for intra-articular deformity [18]. Radiographic measurements to reduce intra-observer variability were repeated by the expert surgeon (LS) after two weeks. Two other surgeons (FB and LB) performed a further evaluation of the radiographic measurements to minimize the inter-observer variability. In case of disagreement, we calculated the mean value between the data obtained. Patients examined were treated with the same posterior stabilized (PS) TKA: JOURNEY™ II Bi-cruciate Stabilized (BCS) Total Knee System (TKS) (Smith & Nephew, Memphis, TN, USA). This implant comes with an anatomical design of the femoral component (medial condyle more distal than lateral condyle, lateral distal condyle less thick than medial femoral condyle, posterior condyles circular in shape) and a PS polyethylene insert with a concave medial surface designed with a medial sulcus near the midline, and a convex lateral surface with a slight posterior slope. Those features are designed to replicate both the PCL and ACL function through range of motion mimicking the physiological rollback of the knee and preventing paradoxical motion. This surgical instrumentation is routinely used at the authors’ institution to treat advanced knee osteoarthritis and is available in our operating room. 

The VERASENSE Knee System was applied in each patient to assess medial and lateral compartment pressure values and to perform soft tissue or bone resection, where necessary, to improve the knee’s final balance. It is a device consisting of microsensors that transmit, through wireless communication, the loading values (lbf.) and the load center of the medial and lateral compartments of the affected knee to a monitor. The measurement data are acquired during the surgical procedure, and thus are monitored through the entire range of motion of the knee (Figure 1 and Figure 2).

All surgical procedures, approved by the authors’ institution, were performed at the CTO Hospital in Turin (Italy) by the same expert knee surgeons. Furthermore, written informed consent was obtained from the patients. 

### 2.1. Surgical Technique

Surgical procedure was performed with the patient in a supine position under general or spinal anaesthesia. A median skin incision was performed in all patients, followed by a medial parapatellar arthrotomy. Bone cuts were then performed using an intramedullary rod for the femur and an extramedullary guide for the tibia. Thanks to this specific implant design, the performed “mechanical” cuts resulted in “anatomical” positioning of the components with 3° physiological joint line on the coronal plane. A 3° external rotation of the femoral component was achieved using an anterior reference guide. The authors tried to avoid excessive external rotation to reduce the risk of instability in mid-flexion [19]. As reported in the literature, an external rotation of 3° has been generally used [20]. After tibial and femoral cuts and removing any osteophytes, a laminar spreader and spacer blocks were used to achieve a symmetrical, rectangular gap in knee flexion and extension. A slightly wider gap on the lateral compartment (no more than 2 mm compared to the medial compartment) was sought during knee varus-valgus stress tests to reproduce the medial pivot design of the implant [21,22,23]. The patella was replaced in each case where there was an apparent patellofemoral arthrosis. The next step was to implant the femoral and tibial component trial based on the bone cuts. Then, the tibial insert trial was placed to assess the knee stability throughout a full range of motion (ROM) with the patella relocated in the femoral trochlea. After trial component insertion throughout full ROM, soft tissue balancing procedures or further bone resections were performed to reach optimal subjective knee balance. Here, the VERASENSE Knee System was inserted into the tibial tray to perform the initial quantitative and objective evaluation of compartment pressures (Figure 3 and Figure 4). After patella reduction and temporary restoration of capsular and medial retinaculum continuity with one or two Backhaus forceps, the surgeon held the leg in a neutral position. Therefore, the side of the affected limb was selected (Figure 5) and both medial and lateral compartment pressure were monitored from full extension to full flexion. As reported by other authors [14,15] who applied the same VERASENSE Knee System sensor model based on the user guide, the ROM was divided into three sections, and the compartmental pressure values when 10°, 45°, 90° of flexion were recorded (Figure 6). Based on this real-time data, adjustments were made to achieve a balanced knee. According to literature, a differential compartment pressure below 15 lbf was considered adequately “balanced” [15,16]. After the initial evaluation, if a mediolateral differential compartment pressure >15 lbf was found, additional soft tissue releases or bone resections were performed and recorded. Following the balancing procedures, the sensor was inserted back into the tibial tray to reassess the compartmental pressures. Thus, the measurements were recorded immediately after the bone resection and implantation of the trial components (phase I), after each additional procedure and after the final implantation with the definitive prosthetic components (phase II). The sensor was reset before insertion into the tibial tray at each stage to minimize errors due to plastic deformation. 

Several balancing procedures were performed in cases where optimal balancing after phase I was not observed. The pie-crusting (PC) technique with a scalpel was used to release soft tissues [24]. When ligament balance was not satisfactory, further bone resection was performed. In most cases, varus resections or other proximal resections of the tibia were performed to achieve a proper knee balance. Three authors (LS, FB and LB) carried out a postoperative radiographic evaluation to detect the correct positioning of the femoral and tibial prosthesis components.

### 2.2. Data Extraction

The senior author (LS) performed a data collection tool helped by other authors (FB and LB). We analysed the following data: age at the time of the surgical procedure, sex, knee side, body mass index (BMI), HKA and JLCA through X-ray analysis, type of balancing procedure performed, medial and lateral compartment pressures recorded with the VERASENSE Knee System sensor immediately after initial bone resections and optimal subjective knee balance after trial component insertion, at the end of additional balancing procedures and after implantation of the definitive components.

### 2.3. Statistical Analysis

All data were analysed using Statistical Product and Service Solutions (SPSS Inc., Chicago, IL, USA). Mean, standard deviation values, and paired *t*-tests were used for continuous data. The Chi-square test was calculated to analyse categorical variables. Linear correlation was performed to examine the relationship between the phase I and phase II compartment pressures differences. A *p*-value < 0.05 was considered statistically significant.

## 3. Results

### Epidemiological and Radiological Data

A total of 21 patients were registered in our study. There was a female predominance (n = 16, 76.19%). The average age of included patients was 77.05 ± 5.51 years (range 68 to 88 years), with a body mass index (BMI) of 27.78 ± 2.82 (range 23.56 to 32.87) at the surgery. Radiographically, the HKA was 7.43° ± 4.14 varus (range −2 to 15°), The JLCA was 1.82 ± 1.6 (range 0 to 5) (Table 1).

Thirteen patients (61.90%) showed a “balanced” knee without the need for further ligament balancing or additional bone resections at the time of measurement, and eight patients (38.10%) showed an intra-operative “unbalanced” knee and required additional procedures. A total of 13 additional balancing procedures (including soft tissue release and bone resections) were performed appropriately. Mainly, five proximal tibia resections (varus and neutral tibial resection) and two posterior slope tibial recut constituted the additional bone procedures; two collateral ligament (MCL) releases by the pie-crusting method, four sub-periosteal superficial MCL releases were performed to achieve a proper balancing. After balancing procedures were performed wherever required, in all patients (n = 21, 100%), a final compartment pressure difference below 15 lbf was recorded in the entire range of motion (Table 2). Higher absolute pressure values were found in the medial compartment than in the lateral compartment in every degree of ROM examined both in the first measurements (phase I) and after implanting the definitive components (phase II). Moreover, pressure values in both compartments showed a trend to decrease with the increase in flexion degrees. The mean values of both medial and lateral compartment pressures were lower in phase II than in phase I (Figure 7). 

In the lateral compartment, the final pressure (phase II) was significantly decreased compared to the initial pressure (phase I) throughout the full range of motion analysed (*p* < 0.05). Moreover, the regression analysis showed a positive linear correlation between phase I and phase II regarding the pressure values of both compartments throughout the entire ROM, with a greater positive correlation when the pressure values of both compartments were examined at 10° of flexion. Consequently, the surgeon could expect reproducible compartment pressure between the trial and final implanted components: this facilitates the prediction of final load measurements during the surgery. The simple linear regression coefficients of the medial compartment were as follows: 10° flexion (R2 = 0.8092; y = 1.6693x − 8.2892), 45° flexion (R2 = 0.6656, y = 1.5143x − 1.7527), 90° flexion (R2 = 0.2384; y = 0.961x + 4.1763). The simple linear regression coefficients of the lateral compartment were as follows: 10° flexion (R2 = 0.6686; y =1.0119x + 3.8807), 45° flexion (R2 = 0.0845, y = 0.529x + 8.892), 90° flexion (R2 = 0.0.1587; y = 0.4924x + 6.1136) (Figure 8). The dynamic movement of the knees in a full ROM was performed based on the user guide of the VERASENSE Knee System to evaluate the joint kinetics. This was done by selecting the Track Button on the VERASENSE Software Application to enable kinetic tracking. The kinetic traces, displayed as green lines, reproduced the medial pivot and femoral rollback, thus reproducing the “original” knee biomechanics as closely as possible (Figure 2).

The postoperative X-ray (anterior-posterior and lateral weight-bearing radiographs) showed a correct implanted component positioning after the surgical procedure. Pre and postoperative x-ray evaluations showed excellent intra-observer and inter-observer coherence. The statistical analysis regarding the prognostic factors evaluated is shown in Table 3. 

A statistically significant correlation between the need for ligament or bone rebalancing and severity of the initial coronal deformity (>10°) was found. Moreover, a statistically significant correlation was seen between the need for ligament or bone rebalancing and a JLCA value >2. There was no statistically significant correlation with an increased requirement for rebalancing after the VERASENSE Knee System was applied for lower deformities. Patients with BMI >30 showed an increase in the need for rebalancing after the VERASENSE Knee System used, but this was not statistically significant. No correlation was found between age and gender and the need for additional ligament or bone balancing after the VERASENSE Knee System was applied. 

## 4. Discussion

This study objectively evaluated compartment pressures of the joint throughout the ROM in TKA designed with a J-curve anatomical femoral design and a PS “medially congruent” polyethylene insert using an instrumented tibial trail. One of the study’s main findings is that the balancing obtained using subjective “feeling” and traditional instrumentation can be inaccurate, resulting in variable results despite extensive experience. Specifically, 61.90% of the patients enrolled showed a “balanced” knee according to Gustke et al. [15] cut-off without the need for further ligament balancing or additional bone resections at the time of measurement, while 38.10% showed an intra-operative “unbalanced” knee and required additional procedures. 

The results obtained are consistent with Wood et al. randomised controlled trial (RCT) in which 35.5% of TKAs performed without sensor-guide were unbalanced [25]. Moreover, as reported by Golladay et al. [26], in a recent prospective multi-center study, not only more balanced knees were observed in sensor-guided TKA than in surgeon-guided TKA (84.0% vs. 50.6%), but also the reported balanced knees were higher in sensor-guided TKA performed by inexperienced users than in surgeon-guided TKA performed by experienced surgeons. 

Those results point out how the sensor technology may be helpful to prevent any incorrect tensioning, ideally preventing joint stiffness, instability, or aseptic loosening, which are some of the main reasons for revision after primary TKA [27]. Specifically, in our series, the objective compartment pressure assessment through the VERASENSE knee system led us to identify and correct the reported imbalance by obtaining a “balanced knee” with a mediolateral pressure difference lower than 15 lbf throughout the ROM in all patients (n = 21, 100%). 

Higher absolute pressures were found in the medial compartment than in the lateral compartment in each ROM degree investigated. Moreover, those pressure values showed a trend to decrease with the increase in flex-ion degrees in both compartments. This compartment pressures path and the TKA design may help restore the medial pivoting movement of the knee and achieve a physio-logic posterolateral rollback of the lateral femoral condyle reported by Risitano et al. [21].

Our study also analysed the “Kinetic Tracking” function of the device: it displays the dynamic motion of the knee through the full ROM to evaluate joint kinetics. The obtained kinetic traces reproduced the knee’s medial pivot and femoral rollback, mimicking natural knee biomechanics (Figure 2). Moreover, as reported by recent studies [7,8], because patients undergoing TKA better appreciate a moderate degree of laxity, especially in the lateral compartment, over higher stiffness, a more reproducible method to intraoperatively assess ligament balancing and rotation of the components may ideally help surgeons to improve clinical results and survivorship of TKA.

Given the results reported in the literature seems reasonable to identify predictor variables that can preoperatively help the surgeon identify the need for sensor-support to focus the indications for the use of those devices. For this purpose, we reported a statistically significant correlation between the need for ligament or bone rebalancing and severity of the initial coronal deformity (>10°) and a preoperative JLCA value >2°.

Those results are consistent with literature [28,29]: acute or late instability is typically related to a preoperative deformity of the knee associated with a persistent or iatrogenic ligamentous asymmetry. Specifically, in the varus knee there is frequently a contracture/shortening of the posteromedial capsule, semimembranosus tendon, and the pes anserinus while in the valgus knee we encounter contracture of the lateral collateral ligament (LCL), iliotibial band (ITB) and lateral capsule accompanied by loosening of medial soft tissues. This bone deformity and soft tissue imbalance frequently results in an increased JLCA and often require a large surgical correction associated with an aggressive ligament release. 

Moreover, patients with BMI >30 showed an increase in the need for rebalancing after the VERASENSE Knee System used, but this was not statistically significant. 

Consequently, the surgeon, evaluating those parameters before surgery, may anticipate difficulties in knee balance and require those devices to achieve the desired result objectively.

There are several limitations to this study. First, it should be noted that the definition used to designate a balanced knee was proposed by Gustke et al. in a non-evidence-based manner relying on previous biomechanical research on passive condylar pressures, intraoperative observations, and the observed significant drop-off in postoperative patient-reported outcome (PROM) scores with intercompartmental loading differences exceeding 20 lb. Despite so, to our knowledge, there are no validated target load in literature to designate a balanced knee, and this is the more commonly used [12,30]. Second, while the pressure values were obtained during surgery, as reported by Bellemans et al., there is evidence that stress relaxation of the ligaments occurs perioperatively and leads to increased ligament laxity [31]. Third, this is a retrospective study enrolling a limited number of patients with no control group; consequently, further analyses will be necessary to determine predictor variables that may recommend those sensor devices in TKA balance.

## 5. Conclusions

Using a sensor during TKA to perform a quantitative evaluation of compartment pressure leads to more reproducible balancing of the operated knee. Surgeons’ ability to objectively balance a knee using conventional instrumentation successfully in about two-thirds of cases in this study. In contrast, when relying on sensor data to balance the knee, a quantitatively balanced knee was obtained in all patients. Moreover, to focus the indications for the use of those devices, we reported a statistically significant correlation between the need for ligament or bone rebalancing and severity of the initial coronal deformity (>10°) and a preoperative JLCA value > 2°. Consequently, the surgeon, evaluating those parameters before surgery, may anticipate difficulties in knee balance and require those devices to achieve the desired result objectively.

## Figures and Tables

**Figure 1 sensors-21-05427-f001:**
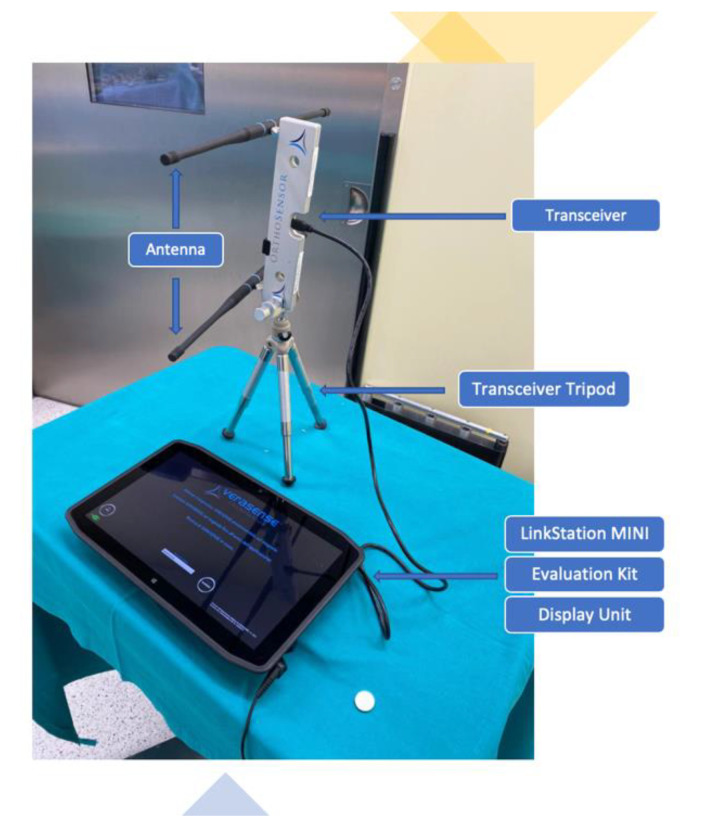
Accessories for the operation of the VERASENSE device: VERASENSE Software Application, LinkStation MINI, Evaluation Kit, Display Unit.

**Figure 2 sensors-21-05427-f002:**
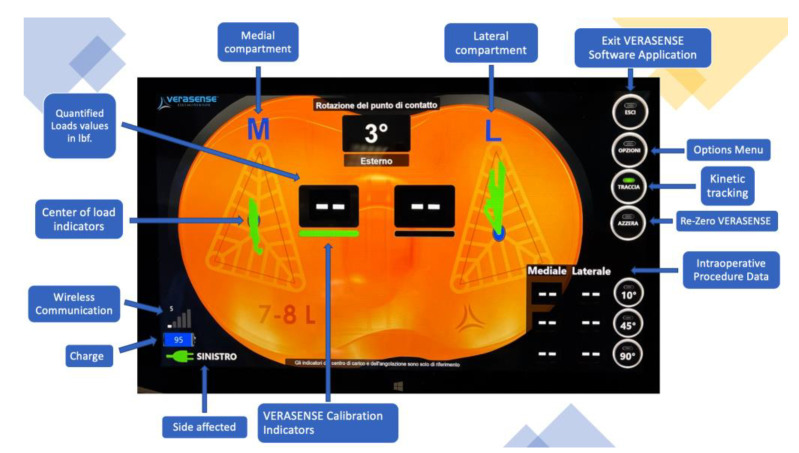
Sensor output representation. The dynamic movement of the knees assessed using Kinetic Tracking. The kinetic traces displayed as green lines reproduce the medial pivot rollback of the knee.

**Figure 3 sensors-21-05427-f003:**
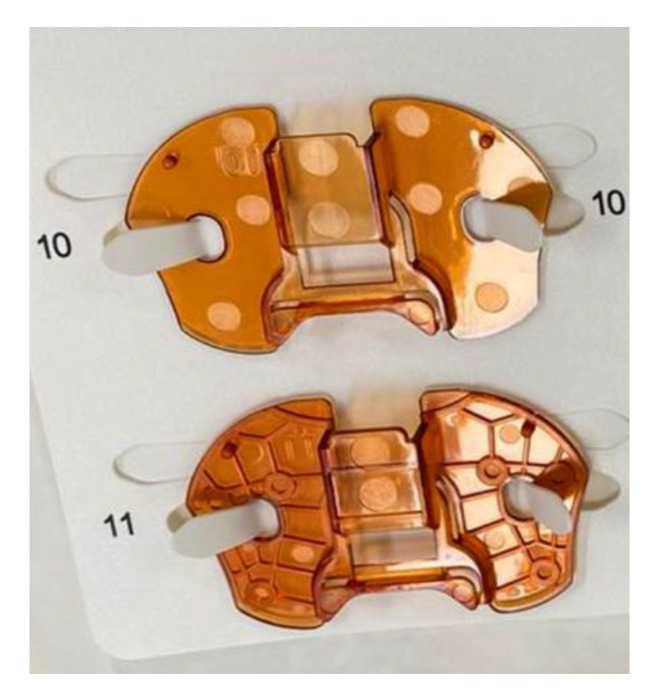
Selection of the most appropriate VERASENSE size.

**Figure 4 sensors-21-05427-f004:**
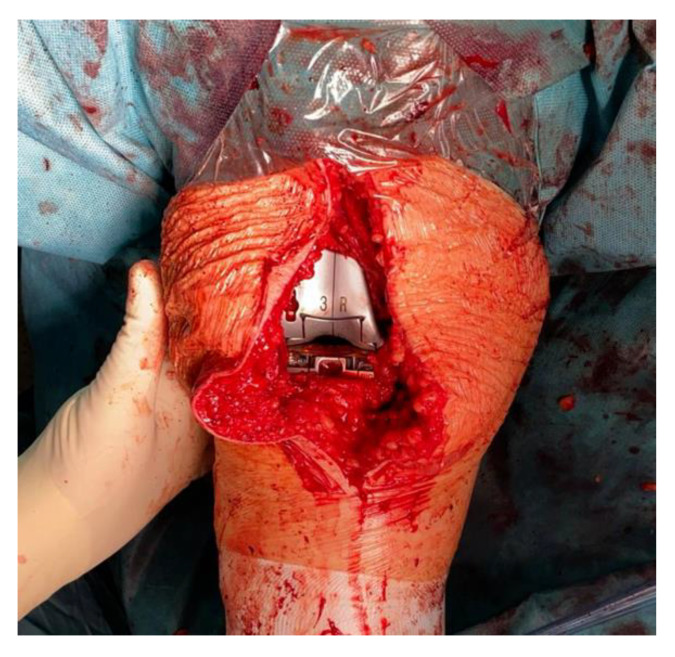
The sensor was inserted into the tibial tray to perform the evaluation of compartment pressures.

**Figure 5 sensors-21-05427-f005:**
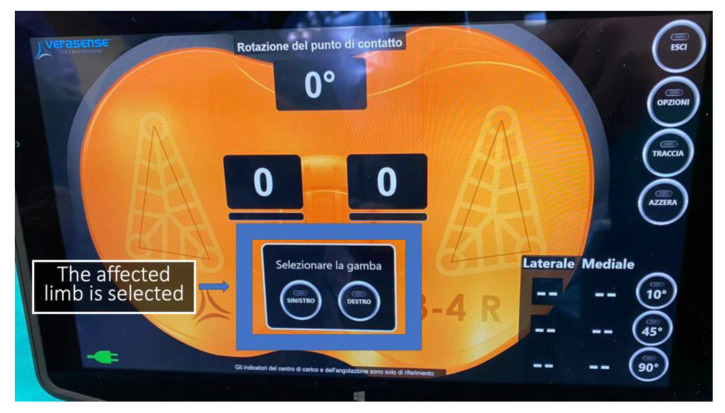
After completing the calibration, the SELECT LEG dialogue appeared. Now select LEFT or RIGHT to assign the side of the knee to be treated.

**Figure 6 sensors-21-05427-f006:**
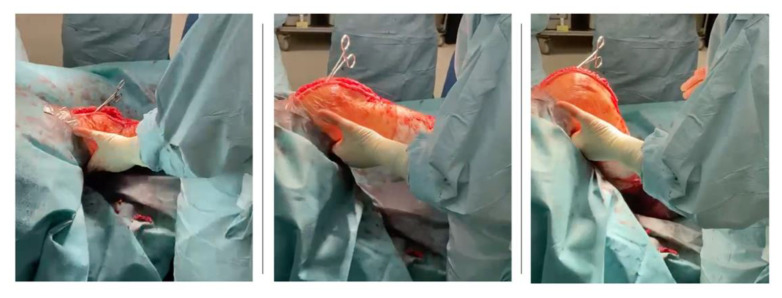
Based on the VERASENSE Knee System sensor model (OrthoSensor Inc., Dania Beach, FL, USA) user guide, the ROM was divided into three sections and the compartmental pressure values at 10°, 45°, 90° of flexion were recorded.

**Figure 7 sensors-21-05427-f007:**
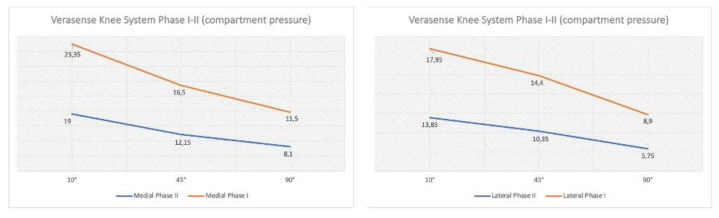
Initial (phase I) and final (phase II) average pressure of the medial and lateral compartments in the ROM degrees evaluated.

**Figure 8 sensors-21-05427-f008:**
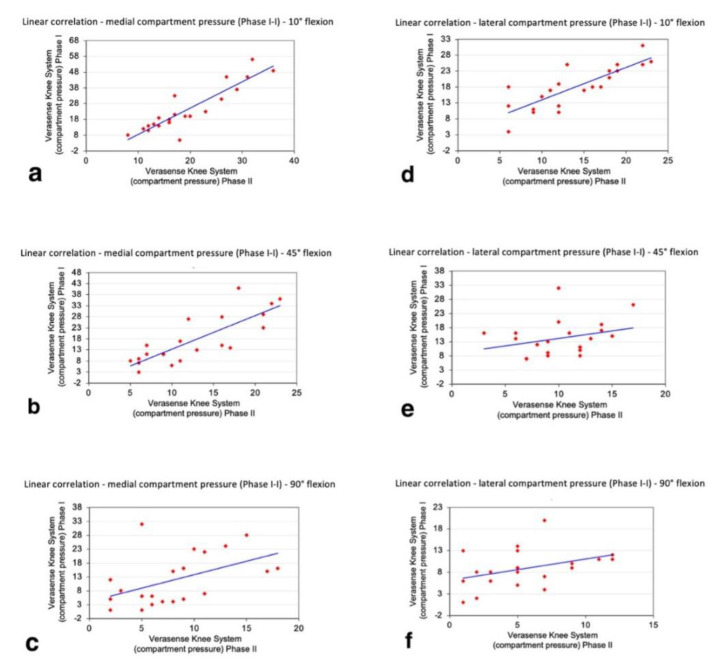
Regression analysis between phase I and phase II and medial and lateral compartment pressures measured (**a**–**f**).

**Table 1 sensors-21-05427-t001:** Epidemiological and radiological data related to additional Balancing Procedures performed. BMI: Body Mass Index, HKA: hip-knee-ankle angle, JLCA: joint line convergence angle, F: female, M: male, R: right, L: left, +: varus angle, −: valgus angle.

Case	Sex	Age	Side Affected	BMI	HKA	JLCA	Balancing Procedure Performed	Type of Balancing Procedure Performed	
								Soft tissues	Bone cuts
1	F	79	R	23.56	+12	5	YES	YES	YES
2	F	77	R	23.88	0	1	NO	NO	NO
3	F	82	L	23.88	+2	1	NO	NO	NO
4	M	72	L	24.76	+2	1	NO	NO	NO
5	F	88	R	24.89	0	4	YES	YES	YES
6	F	86	R	25.39	−1	1	NO	YES	NO
7	F	77	L	25.89	+4	3	YES	NO	YES
8	F	72	L	26.04	−2	1	NO	NO	NO
9	F	82	R	26.67	−1	1	NO	NO	NO
10	M	68	L	27.76	+10	2	YES	NO	YES
11	M	80	R	28.04	+9	1	NO	NO	NO
12	M	77	R	28.3	+1	0	NO	NO	NO
13	F	76	R	28.89	+9	0	NO	NO	NO
14	F	70	R	29.09	+7	3	NO	NO	NO
15	F	76	L	29.41	+11	2	NO	NO	NO
16	F	86	R	30.1	+10	2	YES	NO	YES
17	M	75	L	30.45	+11	4	YES	YES	YES
18	F	76	R	30.86	−2	1	YES	YES	YES
19	F	70	L	31.14	+13	1	YES	YES	NO
20	F	72	L	31.56	+15	1	NO	NO	NO
21	F	77	R	32.87	+8	1	NO	NO	NO

**Table 2 sensors-21-05427-t002:** Prognostic factors related to additional Balancing Procedures performed. Chi-square test. The *p* values of < 0.05 was considered significant. Significant results are reported in bold.

Prognostic Factors	Values	N°	%	Balance Procedure Performed	Balance Procedure not Performed	χ^2^	OR	*p* Value
Sex	F	16	76.19%	6	10			
	M	5	23.81%	2	3	0.01	1.11	0.920
Age	<75 y.o.	7	33.33%	3	4			
	≥75 y.o.	14	66.67%	5	9	0.10	0.74	0.751
BMI	<30	15	71.43%	4	11			
	≥30	6	28.57%	4	2	2.91	5.5	0.088
HKA	<+10 degrees	14	66.67%	3	11			
	≥+10 degrees	7	33.33%	5	2	4.95	9.17	**0.026**
JLCA	<2 degrees	13	61.90%	2	11			
	≥2 degrees	8	38.09%	6	2	7.46	16.50	**0.006**

**Table 3 sensors-21-05427-t003:** Overall Comparison of Phase I and Phase II Compartment Pressure, Paired-t test. The *p* values of <0.05 was considered significant. Significant results are reported in bold.

	Phase I	Phase II	*p*-Value
Medial compartment			
10°	24.38 ± 14.50	19.57 ± 7.81	0.188
45°	17.43 ± 10.97	12.67 ± 5.91	0.088
90°	12.05 ± 9.31	8.19 ± 4.73	0.098
Lateral compartment			
10°	18.10 ± 6.69	14.05 ± 5.41	**0.037**
45°	14.33 ± 6.26	10.29 ± 3.44	**0.013**
90°	8.86 ± 4.33	5.57 ± 3.50	**0.010**

## Data Availability

Data is contained within the article.

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
