# Peer review of "Kinetic Sensors for Ligament Balance and Kinematic Evaluation in Anatomic Bi-Cruciate Stabilized Total Knee Arthroplasty"

_sensors, 2021, doi:10.3390/s21165427_

Round 1
Reviewer 1 Report
The authors examine the use of a knee balancing device during TKA in 21 patients and conclude that this device is satisfactory in improving the balance throughout flexion and conclude that it a useful addition when the coronal deformity was >10 degrees. However the authors exclude valgus deformity >5 degrees as well as patients with RA. Why?
The article is largely about a different balancing device but very little information is given on the device. There needs to be a detailed description (photo) and discussion on how it is used - is it used throughout flexion as other types of balancers are used or can it only be used in the 3 angles described, and why just limit to 10, 45, 90 degrees? What is kinetic tracking and what is it supposed to show? Is there any validation of its use?
Instability is not the leading cause for rTKA, aseptic loosening is the single most common cause in all registries (needs to be altered in the introduction)
What is a bi-cruciate stabilized knee when the PCL is divided? Describe the prosthesis and how it is bi-cruciate.
Finally clinical data is irrelevant in such a small cohort with short follow up and all reference to this should be removed, as well as cost benefit (last comment) as there has been no cost benefit analysis.
Reviewer 2 Report
The research is well conducted and offers some interesting insight on TKA.
Methods are clearly reported and robust, results clear and the discussion well structured.
I have only some minor points the Authors should address:
Abstract
L.29–31. A correlation was found between the compartment pressure of phase I and phase II at 10º of flexion. You should modify the description.
Results
L. 197 Please add 10º flexion.
Discussion
Please consider why a positive correlation was obtained only for 10º of flexion in Fig. 2.
L. 285–293. Please add a more detailed anatomical background on the effect of joint deformities and other factors on rebalancing.
Round 2
Reviewer 1 Report
The authors have addressed my concerns adequately.
However I do question the introduction of the paragraph on causes of knee contracture in the discussion. At best "contracture of the MCL" is a controversial statement with many experienced knee surgeons suggesting that the contracture is capsular and there are no "contractile" cells in the MCL (see work by David Beveland). In my opinion safer to remove this paragraph.